# A Novel Glass Fiber Coated with Sol–Gel Poly-Diphenylsiloxane Sorbent for the On-Line Determination of Toxic Metals Using Flow Injection Column Preconcentration Platform Coupled with Flame Atomic Absorption Spectrometry

**DOI:** 10.3390/molecules26010009

**Published:** 2020-12-22

**Authors:** Eleni Lazaridou, Abuzar Kabir, Kenneth G. Furton, Aristidis Anthemidis

**Affiliations:** 1Laboratory of Analytical Chemistry, Department of Chemistry, Aristotle University, 54124 Thessaloniki, Greece; lazaridoue@chem.auth.gr; 2International Forensic Research Institute, Department of Chemistry and Biochemistry, Florida International University, Miami, FL 33131, USA; akabir@fiu.edu (A.K.); furtonk@fiu.edu (K.G.F.)

**Keywords:** flow injection, atomic absorption spectrometry, automation, solid phase extraction, sol–gel, glass fiber, lead, chromium

## Abstract

A novel simple and sensitive, time-based flow injection solid phase extraction system was developed for the automated determination of metals at low concentration. The potential of the proposed scheme, coupled with flame atomic absorption spectrometry (FAAS), was demonstrated for trace lead and chromium(VI) determination in environmental water samples. The method, which was based on a new sorptive extraction system, consisted of a microcolumn packed with glass fiber coated with sol–gel poly (diphenylsiloxane) (sol–gel PDPS), which is presented here for the first time. The analytical procedure involves the on-line chelate complex formation of target species with ammonium pyrrolidine dithiocarbamate (APDC), retention onto the hydrophobic sol–gel sorbent coated surface of glass fibers, and finally elution with methyl isobutyl ketone prior to atomization. All main chemical and hydrodynamic factors, which affect the complex formation, retention, and elution of the metal, were optimized thoroughly. Furthermore, the tolerance to potential interfering ions appearing in environmental samples was also explored. Enhancement factors of 215 and 70, detection limits (3 s) of 1.1 μg·L^−1^ and 1.2 μg·L^−1^, and relative standard deviations (RSD) of 3.0% (at 20.0 μg·L^−1^) and 3.2% (at 20.0 μg·L^−1^) were obtained for lead and chromium(VI), respec tively, for 120 s preconcentration time. The trueness of the developed method was estimated by analyzing certified reference materials and spiked environmental water samples.

## 1. Introduction

Flow injection and related techniques have been proved to be suitable for on-line fluidic manipulation as well as for the successful sample processing in an automated operation mode, meeting the standards of green analytical chemistry [1,2]. Within the last decades, on-line sorbent extraction is by far the most utilized sample preparation technique in flow systems due to its straightforward operation, high preconcentration and separation efficiency, versatility and miniaturization ability, as well as advent of advanced sorptive materials such as the sol–gel-based ones. Most often, on-line microcolumns, packed with an appropriate sorptive material, within a flow network, are employed as an efficient front end to atomic spectrometric techniques such as the flame atomic absorption spectrometry (FAAS) or other detection systems, improving significantly the analytical performance characteristics [3]. A plethora of novel sorbents with desirable characteristics such as fast kinetics, physical chemical and mechanical stability, low cost, and availability have been proposed in on-line sorptive extraction methods for toxic metal determination in various types of samples matrices [4,5,6]. Generally, sorbents can be categorized by the nature of primary interaction mechanism with the target analyte as hydrophobic (reversed phase), polar, and ion-exchange.

Sol–gel technology has been proved to be the vehicle for preparing advanced hybrid inorganic–organic polymer coatings with tunable porosity, selectivity, and noticeably improved chemical and thermal stability. The sol–gel process allows chemical integration of the sol–gel sorbent to the substrate surface in diverse forms, such as particles, fiber, fabric, tube capillary, film, and monoliths that can be used in sample preparation micro-techniques [7,8]. Malik and co-workers reported for the first time the sol–gel process for solid phase microextraction (SPME) using hydroxy-terminated polydimethylsiloxane (PDMS) to create surface-bonded organic inorganic hybrid coating for SPME fiber [9]. The sol–gel PDMS-coated fiber offers better extraction selectivity than the commercially available one and was found to be efficient in extracting both polar and nonpolar analytes from aqueous samples. Due to the chemical bonding and the open porous structure of the sol–gel PDMS coatings, they provide comparable or better extraction efficiency with a shorter equilibrium time and significantly higher thermal stability [7]. The application of SPME fiber coated with sol–gel sorbent for extracting organometals from aqueous samples followed by HPLC analysis was first proposed by Gbatu et al. [10] while, up until today, the sol–gel sorbents for extracting metals and organometallics are used only as in-tube or capillary surface coatings, prior to inductively coupled plasma mass spectrometry (ICP-MS) determination [11]. A comprehensive review of SPME for trace elements speciation was published by Mester et al. [12].

In 2014, Kabir and Furton presented [13,14] a novel green sample pretreatment technique, fabric phase sorptive extraction (FPSE) overcoming the majority of the problems often encountered in modern sample preparation approaches such as low sorbent capacity and long sample preparation time [15]. The inherent porous surface of a natural or synthetic fabric substrate such as cotton, cellulose, polyester, and glass fiber together with the strength of sol–gel derived hybrid sorbents, which are uniformly dispersed in the form of an ultra-thin film into the fabric, result in a plethora of sorbents with significant retention capacity and satisfactorily fast extraction equilibrium to carry out the extraction of the analyte. Recently, Anthemidis et al. reported [16] for the first time the automation of the FPSE technique for on-line extraction and the determination of toxic metals in environmental samples by FAAS. The proposed fabric disk sorptive extraction (FDSE) technique is based on firmly packing an adequate number of fabric disks, obtained from a sol–gel sorbent-coated fabric, into a microcolumn incorporated in an on-line flow injection (FI) system.

In the frame of evaluation of new sol–gel sorbents in various geometrical shapes, Anthemidis et al. developed a novel green automatic sample preparation approach called fabric fiber sorbent extraction (FFSE) [17]. FFSE is based on the effective packing of sol–gel coated polyester fibers, into a FI on-line microcolumn, which operates as a front end to FAAS or other detectors for metal or organic analytes determination. Taking into consideration the advantages of sol–gel technology in producing the chemically bonded surface of fiber, a new category of sol–gel sorbents is introduced, resulting in a plethora of diverse fiber coatings, which potentially can be packed into an on-line preconcentration mini column for automated flow injection systems.

In the present work, a new sample separation and/or preconcentration platform named fiber coated sorbent extraction (FCSE) in which a glass fiber substrate coated with sol–gel poly (diphenylsiloxane) (sol–gel PDPS) was prepared and utilized for the first time as packing material for on-line microcolumn preconcentration systems coupled with FAAS for metal determination. Lead and chromium were chosen as model metals for the development and evaluation of the proposed method considering their toxicity even at very low concentrations. The applicability of three sol–gel materials, sol–gel poly (diphenylsiloxane) (sol–gel PDPS), sol–gel poly (dimethylsiloxane) (sol–gel PDMS), and sol–gel methyltrimethoxysilane (sol–gel MTMS), coated on glass fibers and cotton balls packed in microcolumns was further investigated. The proposed method was optimized for all parameters and the accuracy was evaluated by analyzing certified reference materials and real samples.

## 2. Materials and Methods

### 2.1. Reagents Materials and Samples

Analytical grade chemical reagents were used and provided by Merck (Darmstadt, Germany, http://www.merck.de) and ultra-pure quality water purification system produced by Milli-Q (Millipore, Bedford, TX, USA, http://www.millipore.com) was used all over the study. Lead(II) and chromium(VI) working standard solutions were prepared by appropriate stepwise dilution from a 1000 mg·L^−1^ in 0.5 mol·L^−1^ HNO_3_ stock standard solutions (Titrisol, Merck) immediately prior to use. The standard solutions of Pb(II) and Cr(VI) as well as the samples were prepared in 0.01 mol·L^−1^ HNO_3_ (pH ≈ 2.0). The chelating reagent solution of ammonium pyrrolidine dithiocarbamate, APDC and ammonium diethyl dithiophosphate, DDPA (Aldrich, Steinheim, Germany), at 0.05% (*m*/*v*) concentration level were prepared daily by dissolving the appropriate amount of solid into water. Methyl isobutyl ketone (MIBK) was used after saturation with water. Laboratory glassware was rinsed with water after decontaminating overnight in a 10% (*v*/*v*) nitric acid solution. 

Three certified reference materials (CRMs) were used in order to verify the accuracy of the developed method; NIST CRM 1643e (National Institute of Standard and Technology, Gaithersburg, MD, USA) containing trace elements in water, BCR 278-R (Community Bureau of Reference Brussels, Belgium) containing trace elements in mussel tissue, for lead determination and NIST SRM 2109 for Cr(VI) determination. The SRM 2109 is a standard reference solution with a certified Cr(VI) content of 1000 mg·L^−1^, which was properly diluted to a final Cr(VI) concentration of 20.0 μg·L^−1^.

Environmental water samples were collected from sampling sites located in Northern Greece during September 2020: Axios river, Volvi lake, Toroneos gulf (Chalkidiki). All samples were filtered through 0.45 μm membrane filters, acidified to ca. pH 2 with dilute nitric acid and stored at 4 °C in acid-cleaned polyethylene bottles. These solutions were used for the determination of the “dissolved” fraction of the metal in the samples.

### 2.2. Apparatus

The detection system consisted of a Perkin-Elmer model 5100 PC flame atomic absorption spectrometer (Norwalk, CT, USA, http://las.perkinelmer.com). A lead electrodeless discharge lamp (EDL), operated at 10 W and a chromium hollow cathode lamp (HCL) operated at 30 mA were used as light sources. The resonance lines were set at 283.3 nm 357.9 nm for lead and chromium respectively and the monochromator spectral bandpass (slit) was fixed at 0.7 nm. The flame composition was optimized in order to compensate the effect of methyl isobutyl ketone (MIBK) contribution to the flame, which serves as additional fuel. The flow rate of air and acetylene was set at 10.0 L·min^−1^ and 0.9 L·min^−1^ so as to obtain an oxidizing flame for lead and at 10.0 mL·min^−1^ and 4.0 mL·min^−1^ to obtain a reducing flame (yellow rich) for chromium determination. The resulting nebulizer free uptake rate was measured to be 4.4 mL·min^−1^. The spray chamber was equipped with an internal PTFE flow spoiler for better nebulization conditions, according to the manufacturer’s recommendations. 

A Perkin-Elmer (Norwalk, CT, USA) Model FIAS-400 flow injection system operated in preconcentration mode was coupled with the 5100 PC FAAS. The FIAS-400 manifold comprised a peristaltic pump for sample or standard solution and chelating reagent propulsion via tygon tubing and a micro-syringe pump (MicroCSP-3000, FIAlab instruments, Bellevue, WA, USA) equipped with a 2.5 mL glass barrel (TECAN) and a three-position Teflon/Kel-F valve on the top of it, for MIBK (eluent) propulsion. The FI system includes a 5-port 2-position injection valve, IV, which integrates the microcolumn, C fixed at ports 2 and 4, as it is shown in Figure 1 (Appendix A). The integrated FAAS/FIAS-400 system was controlled by a personal computer operated with the AA Lab Benchtop version 7.2 software program. The connecting tube between the IV and the nebulizer of FAAS was as short as possible, (polytetrafluoroethylene (PTFE) tube, 20 cm length, 0.5 mm i.d.). A VICI^®^ (Valco Instruments Co. Inc.) (Houston, TX, USA) three-section “T” type confluence connector made of polyether ether ketone (PEEK) with 0.5 mm i.d. bore size, serving as a flow compensation (FC) unit, was adapted between the FAAS nebulizer and the FIAS system. All the tubing of the flow system was made of polytetrafluoroethylene (PTFE).

### 2.3. Constraction of the FCSE Microcolumn 

The microcolumns were constructed from a plastic (polypropylene) 1.0 mL disposable syringe (length ≈ 12 cm; i.d. 5.0 mm) with a luer slip centered tip. The main body of syringe was cut properly at a length ca. 40 mm., and the obtained packed column had an effective length of 30 mm and i.d. of 5.0 mm. At first, the glass fibers were cut in small pieces (ca. 1–2 mm) using scissors Appendix A, so that they could be packed in the column. An amount of 200 mg of glass fiber coated with sol–gel PDPS was firmly packed into the column, as it is shown in Appendix A). It was not necessary to establish frits at the two ends of the column for glass fiber immobilization. Two push-fit connections made of silicon plug and short PTFE tubing (i.d. ≈ 0.5 mm) were used in order be installed in the FI manifold, at ports 2 and 4 (Figure 1 and Appendix A). 

This arrangement facilitates easy and constant flow between the glass fibers and effective contact with the coating surface. Therefore, a high loading flow rate can be adopted for high preconcentration ratio and short analytical cycles. The glass fibers are packed into the column in a chaotic way, which prevents the formation of canals through the mass of sorbent. Before the application of a fresh column in the FI system, it is flushed twice with 1000 μL of MIBK and deionized water subsequently. As it was proved from the present study, the extraction performance and the retention efficiency were unchanged as well as the packing material remained stable for more than 500 loading/elution cycles for the two metals.

### 2.4. Preparation of Glass Giber and Cotton Ball Coated with Sol–Gel Sorbents 

The aim of the present work was to study various substrates in the form of fibers coated with diverse sol–gel sorbent materials, as packing sorbents in on-line columns either for hydrophobic or hydrophilic analytes. For this purpose, two fiber types were used; the glass fiber (GF) and the cotton ball (CB), coated with sol–gel poly (diphenylsiloxane) (sol–gel PDPS), sol–gel poly (dimethylsiloxane) (sol–gel PDMS), and sol–gel methyltrimethoxysilane (sol–gel MTMS).

The glass fiber is made from fine glass fibers with individual filament diameters of approximately 0.008 mm. The glass fiber is pliable and resistant to breakage. The borosilicate glass used for preparing the glass fiber contains minimal heavy metals—an attribute that is advantageous for heavy metal extraction and analysis. The glass fiber contains a high concentration of surface Si-OH functional groups that can be easily exploited to anchor with the ligand/polymer of choice to create a new microextraction system via sol–gel sorbent coating technology. The cotton ball is made from 100% cotton cellulose possessing hydrophilic surface property due to the presence of accessible C-OH functionality. These carbinol functional groups can be used to chemically bind different polymer/ligands via the sol–gel process to create a new microextraction system.

For an effective sol–gel sorbent, the selection of the sol–gel active organic polymer, the inorganic or organically modified inorganic sol–gel precursor (that forms a link between the substrate and the organic polymer), the solvent/solvent system, the catalyst, the amount of water, as well as an appropriate relative molar ratio of the constituents must be considered.

Prior to the sol–gel sorbent coating on the substrates, the segments of fiber substrates (e.g., glass, cotton) are first soaked with deionized water under sonication in order to become thoroughly wet. Then, fibers are cleaned with a high amount of deionized water so that chemical residues are removed. After, a process called mercerization follows by treating the fibers with 1.0 mol·L^−1^ NaOH under sonication, and the mercerized fiber is washed several times with plenty of deionized water. The next step is treating the fiber with 0.1 mol·L^−1^ HCl under sonication, washing again with deionized water, and finally drying overnight in an inert atmosphere. Dried GF and CB substrates are stored in clean glass airtight containers until they are coated with the appropriate sol–gel sorbent. The sol solutions for sol–gel sorbent coating were prepared by the sequential addition and subsequent vortexing of the ingredients: organic polymer, methyltrimethoxysilane as the sol–gel precursor, methylene chloride as the solvent, water as the hydrolytic reagent, and trifluoroacetic acid as the acid catalyst at a molar ratio for sol–gel PDPS 0.11: 1: 4.68: 3: 0.12, for sol–gel PDMS 0.004: 1: 4.68: 3: 0.12 and for sol–gel MTMS 0: 1: 4.68: 3: 0.12, respectively. In a sol–gel MTMS sorbent coating, no organic polymer was added to assess the role of organic polymer in the overall selectivity and extraction affinity of the composite sorbent. 

The coating procedure is integrated by inserting gently the treated glass fibers/cotton ball into the vial containing the sol solution. As a result, a 3D network is formed throughout the porous substrate matrix. After a predetermined coating time, the glass fiber/cotton ball is taken away from the sol solution, dried, and placed in a desiccator overnight for solvent evaporation and for aging of the sol–gel coating. The objective of this step is to complete the condensation reaction and remove solvents and unreacted residuals from the sol–gel matrix, ensuring a clean, surface bonded sol–gel sorbent free of structural deformation and internal stress. Then, the coated glass fiber/cotton ball media is rinsed with the appropriate solvent system under sonication for a few minutes in order to remove residual sol solution ingredients from the coated surface.

### 2.5. Automatic On-Line Operational Procedure

The automatic on-line FI-FCSE-FAAS analytical procedure for metal determination is operated in two main steps, specifically, sample loading and the elution step, which are presented schematically in Figure 1, while the operational sequences of the developed method are summarized in Table 1. The developed manifold is operated in time-based mode regarding the volume of sample, chelating reagent and eluent. Each analytical cycle involves two steps, as described below.

In the first step (Figure 1a), the injection valve (IV) is actuated in “loading position”, and the peristaltic pump (P) is activated for the propulsion of the sample or standard solution (flow rate: 8.0 mL·min^−1^) and the chelating reagent, APDC (flow rate: 1.0 mL·min^−1^). The two streams are merged and the resulted mixed stream, which contains the on-line formed metal–APDC complex, is delivered through the FCSE column for an appropriate time (60 or 120 s). This time expresses the preconcentration time and defines the loading sample volume. The metal–APDC complex is retained onto the sol–gel glass fiber sorbent on the way to waste. During this step, a syringe pump (SP) is activated in order to aspirate (1500 μL) MIBK into the syringe barrel (ready for elution), while in the meantime, the FAAS nebulizer aspirates air through the perpendicular inlet of the flow compensation (FC) adapter. After the preconcentration time, in the second step, the pump P is turned off (no sample and APDC is delivered), and the IV is actuated in the “elution position”. MIBK (1500 μL) is dispensed at flow rate 3.0 mL·min^−1^ by the SP through the FCSE column in order to elute the analytes and deliver them to the FAAS atomizer for quantification. It is noteworthy that the eluent flows through the microcolumn in the reverse direction than that of the sample/reagent in order to minimize the dispersion of the analytes into the eluent segment. No extra washing step of the column was necessary, as the recorded signal returned at the baseline level. The recorded peak height signal was proportional to metal concentration in the sample solution. Five replicate measurements per sample were performed in all instances. 

## 3. Results and Discussion

### 3.1. Characterization of Sol–Gel PDPS Coated Glass Fiber

#### 3.1.1. Fourier Transform Infrared Spectroscopy (FT-IR)

The sol–gel PDPS-coated glass fiber was characterized using Fourier Transform Infrared Spectroscopy (FT-IR). Figure 2 represents the FT-IR spectra of (a) pristine PDPS polymer; and (b) sol–gel PDPS-coated glass fiber.

The characteristic peaks of pristine PDPS polymer appear at 694 cm^−1^ and 717 cm^−1^ representing vibration bands for Ph-Si-Ph. The very strong band at 1011 cm^−1^ represents the vibration band for Si-O-Si [18]. Bands at 1591 cm^−1^ and 1428 cm^−1^ can be assigned to skeleton vibration of phenyl groups in PDPS polymer [19]. The shoulder band at 1009 cm^−1^ originates from SiO_2_ network formation, and the appearance of a majority of bands from the PDPS spectra in the sol–gel PDPS coated fiberglass spectra is evidence of the successful integration of PDPS polymer into the sol–gel PDPS network coated on the glass fiber substrate.

#### 3.1.2. Scanning Electron Microscopy (SEM)

The surface morphology of the uncoated glass fiber and sol–gel PDPS glass fibers were investigated using scanning electron microscopy. As can be seen in the SEM images presented in Figure 3, the surface of the sol–gel PDPS coated glass fiber (Figure 3b) is distinctly different than that of the uncoated glass fiber (Figure 3a). The sol–gel PDPS coating on glass fiber looks homogeneous with a characteristic roughened surface. The magnified SEM image of the sol–gel PDPS coated glass fiber (Figure 3c) demonstrated uniform surface coating, thanks to the chemical coating process exploited in sol–gel sorbent coating technology.

### 3.2. Sol–Gel PDPS Coated Glass Fiber and Chelating Agent

Although the chelating agent ammonium pyrrolidine dithiocarbamate (APDC) is a highly polar chelating agent with logK_ow_ < 1.10, after complexing with metal ions, the electron-rich center of APDC becomes overall electronically charge-neutral and therefore, the complex turns relatively hydrophobic. Taking this observation into consideration, a new fiber substrate, glass-fiber was selected along with a non-polar organic polymer, poly-diphenylsiloxane (PDPS) and a methyl terminated trimethoxysilane (MTMS) inorganic precursor in order to create the robust hydrophobic extraction media. Due to the strong covalent bond between the glass fiber substrate and the sol–gel PDPS sorbent network, various organic solvents as well as acidic or alkaline solutions can be used for the effective elution of the extracted analytes without risking the potential loss of the extracting sorbent.

### 3.3. Optimization of Chemical and Hydrodynamic Parameters

All parameters affecting the performance of the FI-FCSE-FAAS method were studied and optimized using the univariate methodology, one variable at a time. The experiments were carried out with sol–gel PDPS coated glass fiber, as it presented the best extraction and operated features, using standard aqueous solutions of 100.0 μg·L^−1^ Pb(II) and 100.0 μg·L^−1^ Cr(VI) for a 60 s preconcentration time.

Regarding the elution, organic solvents such as MIBK, ethanol, and methanol are well established as more effective eluents than acidic or alkaline solutions, in on-line column preconcentration systems, which use hydrophobic sorbents. Among them, MIBK is immiscible with water, less of a polar solvent than alcohols, and contributes positively to the atomization procedure. MIBK produced higher and the sharpest signals, with stable baseline, as it has been proven elsewhere [14]. Therefore, MIBK was adopted for elution, and its volume was fixed at 1500 μL for the complete elution and cleaning of the FCSE column. 

#### 3.3.1. Chelating Agent and Its Concentration

The most commonly used chelating agents in liquid–liquid extraction and solid phase extraction methods for metal determination are the dithiocarbamates such as APDC and dithiophosphates such as ammonium diethyl dithiophosphate (DDPA), which form strong hydrophobic complexes in acidic solutions [20,21]. Preliminary experiments for Pb(II) determination under the proposed FI-FCSE-FAAS method have shown 35% higher sensitivity using APDC than DDPA. The extraction of Cr(III) using APDC as a ligand under the conditions usually employed for the extraction of Cr(VI) and other metals has been found to be inefficient due to the difficulty of displacing the coordinated water from the strongly hydrated Cr(III) ion by the APDC [22]. On contrast, Cr(VI) is reduced easily and quickly by APDC to Cr(III), which subsequently forms chelate and easily gets extracted. This differentiation on complex formation of Cr(III) and Cr(VI) with APDC can be easily applied in on-line preconcentration systems for chromium speciation [23]. In addition, DDPA cannot reduce Cr(VI) in order to form a complex with Cr(III), as it is proved experimentally. Considering the above remarks, APDC was adopted for further studies of both metals. 

The influence of APDC concentration on the absorbance was studied in the range 0.05–0.10% m/v. In the absence of APDC, no extraction was observed for both metals. As it is shown in Figure 4, the absorbance was leveled off at 0.05% for Pb(II) and 0.10% *m*/*v* for Cr(VI). Therefore, a concentration of 0.05% *m*/*v* APDC in water was selected for further experiments as it was proved adequate for the complexation of all analytes.

#### 3.3.2. Effect of the Acidity on the Complexation

It is evidenced that the acidity of the sample solution affects significantly the complexation of the metals with dithiocarbamates and their hydrophobicity, which is necessary for the retention of the metal–APDC complex on a hydrophobic surface such as sol–gel PDPS-coated fiber glass. The effect of sample acidity on the absorbance was studied by varying the nitric acid concentration from 3.0 × 10^−5^ to 5.0 × 10^−1^ mol·L^−1^. As it is shown in Figure 5, maximum absorbance was observed from 0.001 to 0.03 mol·L^−1^ for Pb(II) and from 0.01 to 0.1 mol·L^−1^ for Cr(VI). Thus, the sample solution was fixed at concentration of 0.01 mol·L^−1^ HNO_3_ (pH ≈ 2) for further study.

This is a real advantage of the use of APDC as a chelating agent, since a tight control of pH or a buffer solution is not required, reducing the risk of contamination. In addition, it is worth noting that samples submitted to preservation are usually in an acid solution or acidified with 1% (*v*/*v*) HNO_3_ such as seawater samples, meaning that they are ready for direct analysis [20].

#### 3.3.3. Effect of Loading and Elution Flow Rate

The time-based, on-line preconcentration systems are affected both by the sample loading flow rate and the rate of the complex formation. The effect of the sample flow rate was studied in the range 5.0–9.5 mL·min^−1^, at fixed sample/APDC flow rate ratio of ca. 8.0. The absorbance was found to be increased practically linearly up to 8.0 mL·min^−1^ showing that the kinetic of the complex formation is fast and the contact time is enough (Appendix A). For higher flow rates, the absorbance was increased by a lower ratio. Considering the sensitivity of the method and the sample consumption, a flow rate of 8.0 mL·min^−1^ was selected as optimum for subsequent studies.

The effect of MIBK flow rate was examined within the range of 1.8–4.8 mL·min^−1^ (30–80 μL·s^−1^). Maximum absorbance was recorded at flow rates between 2.4 and 3.6 mL·min^−1^, while at lower and higher flow rates, a decrease was noticed in the analytical signal (Appendix A), which is attributed to the analyte dispersion into the eluent segment during its transportation toward the FAAS nebulizer and to the insufficient elution at higher flow rates. A MIBK flow rate of 3.0 mL·min^−1^ (50 μL·s^−1^) was adopted in further experiments.

#### 3.3.4. Effect of Preconcentration Time

A key factor for on-line time-based preconcentration systems is the preconcentration time, which is defined by the loading time. The preconcentration time affects the main parameters of the method, such as the time of analysis as well as the preconcentration ratio. The effect of the loading time was examined from 30 to 180 s. The resulting signals demonstrated an increasing and almost linear absorbance as the loading time was increased for both metals (Appendix A). Hence, a diverse preconcentration time can be used in order to obtain adequate sensitivity with lower sample consumption. 

### 3.4. Effect of Co-Existing Ions

Ammonium pyrrolidine dithiocarbamate is a chelating agent that has been used for the complexation of several transition metal ions through various extraction techniques. Although atomic absorption spectrometry in principal has an inherent tolerance to interferences arising from potentially coexisting ions, these should be considered and examined. The effect of various coexisting ions occurring in environmental water samples on the recovery of 40.0 μg·L^−1^ Pb(II) and 50.0 μg·L^−1^ Cr(VI) were tested with individual interferents added, using the optimized FI-FCSE-FAAS system for 60 s preconcentration time. Considering as interference a deviation of the recovery more than ± 5%, the obtained results showed that the tolerance concentrations for each studied ion were the following. Al(III), Cr(III), Fe(III), and Mn(II) are tolerated at least up to 5.0 mg·L^−1^; Co(II), Cu(II), and Zn(II) are tolerated at least up to 1.0 mg·L^−1^; and Cd(II) and Hg(II) are tolerated at least up to 0.5 mg L^−1^ for both metals, while Ni(II) is tolerated at least up to 2.0 mg·L^−1^ for Pb(II) and up to 0.5 mg·L^−1^ for Cr(VI). For individual lead determination, Cr(VI) is tolerated up to 5.0 mg·L^−1^, whereas for chromium(II) determination, Pb(II) is tolerated up to 2.0 mg·L^−1^. Alkali and alkaline earth metals at high concentrations, which are present in natural waters, were tested. Na^+^ and K^+^, Ca^2+^, Mg^2+^ and Ba^2+^ are tolerated at least up to 600 mg·L^−1^, while NaCl up to 30 g·L^−1^ did not cause any interference.

### 3.5. Figures of Merit

The analytical performance data of the proposed method FI-FCSE-FAAS, under the optimum conditions for lead and chromium(VI) determination are summarized in Table 2. For a preconcentration time of 60 s and 120 s, the sampling frequency is 40 and 30 cycles per hour (h^−1^), respectively. The detection and quantification limit are calculated by 3s and 10s criterion respectively, according to IUPAC recommendation, as 3 or 10 times the standard deviation of the blank solution measurements (*n* = 10) divided by the slope of the corresponding calibration equation.

By the typical suction of aqueous standard solutions into the FAAS nebulizer, without any preconcentration process, the regression equation was *A* = 0.093 [Pb(II)] + 0.002 and *A* = 0.0023 [Cr(VI)] + 0.001 ([M] in mg·L^−1^, *n* = 5) for lead and chromium(VI) determination, respectively. The enhancement factor was calculated by the ratio of the slopes of calibration curves with and without preconcentration.

A comparative study between the hydrophobic sol–gel PDPS coated on glass fiber and the hydrophobic, sol–gel PDMS and sol–gel MTMS coated on glass fiber under identical conditions for on-line FI-FCSE-FAAS determination of lead was performed with the developed system. The sol–gel PDPS coated glass fiber revealed significant higher sensitivity and better precision against the other coatings, which was due to the weak retention onto the sorbent surface. In addition, the above sol–gel coatings were examined on cotton ball. The results showed that cotton ball cannot be used as packing material for on-line column preconcentration due to the gradual tightening of cotton fibers in the column resulting in high backpressure. This problem is probably arising due to the thinner fibers and the hydrophilic nature of cotton.

The analytical performance data of the proposed method together with that of other on-line solid phase extraction methods for lead and chromium(VI) determination with FAAS, reported in the recent literature [16,17,24,25,26,27,28,29,30,31,32,33,34,35], are given in Table 3. The developed method features satisfactory sensitivity and precision (relative standard deviations, or RSD) with better detection limits (except [24]) and similar precision (RSD) for the lead determination. For chromium(VI) determination, comparable detection limits and precision have been observed with a lower preconcentration time (except [29,30]). 

The accuracy of the FI-FCSE-FAAS method was estimated by the analysis of a CRM 1643e and BCR 278–R for lead and the SRM 2109 for chromium(VI), while the trueness of the method was demonstrated by the Student *t*-test. The obtained analytical results and the *t*_exp_ values are presented in Table 4. Considering that the *t*_exp_ values are lower than the *t*_crit, 95%_ = 4.30, no statistically significant differences were found at the 95% probability level, showing that the method can be applied for the analysis of such types of samples.

### 3.6. Applications in Spiked Environmental Water Samples

The proposed method for preconcentration time of 120 s has been applied to the analysis of natural water samples collected from the Northern Greece area, namely Axios river, Volvi lake and Toroneos gulf (Chalkidiki). The results are given in Table 5. The obtained recoveries were varied within the range 93.0–108.0%, indicating the applicability of the method for lead and chromium(VI) determination in natural water samples.

## 4. Conclusions

A novel automated sample preconcentration and separation platform as a front end to FAAS for trace metal determination has been presented. The system is based on an on-line microcolumn packed with a novel sorbent, glass fiber coated with sol–gel PDPS. As far as we know, the proposed sorbent, sol–gel PDPS-coated glass fiber has not been used for metal or other analyte determination neither in batch nor in automated preconcentration systems. The fiber substrate packed in on-line columns offers limited flow resistance with excellent packing reproducibility, high extraction efficiency, and satisfactory sensitivity. The hydrophobic sol–gel PDPS coated on glass fiber provides good extraction efficiency, especially for metal–APDC complexes, which appear to have low polarity. The proposed automated FI-FCSE-FAAS method proved to be facile, rapid, and accurate for the direct analysis of environmental water samples, even of seawater ones, for lead and chromium determination. It is noted that the new absorbent packaging material highlights the excellent characteristics of glass fibers in combination with the very promising sol–gel chemistry. The proposed manifold is very convenient for a plethora of inorganic or organic analyte determination.

## Figures and Tables

**Figure 1 molecules-26-00009-f001:**
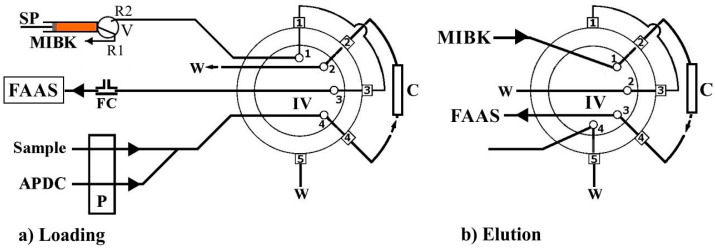
Schematic diagram of the FI-FCSE-FAAS method for Pb(II) and Cr(VI) determination. IV; injection valve in (**a**) loading and (**b**) elution position; SP, syringe pump; P, peristaltic pump; C, fiber coated sorbent extraction (FCSE) column; FC, flow compensation; FI, flow injection; FAAS, flame atomic absorption spectrometry.

**Figure 2 molecules-26-00009-f002:**
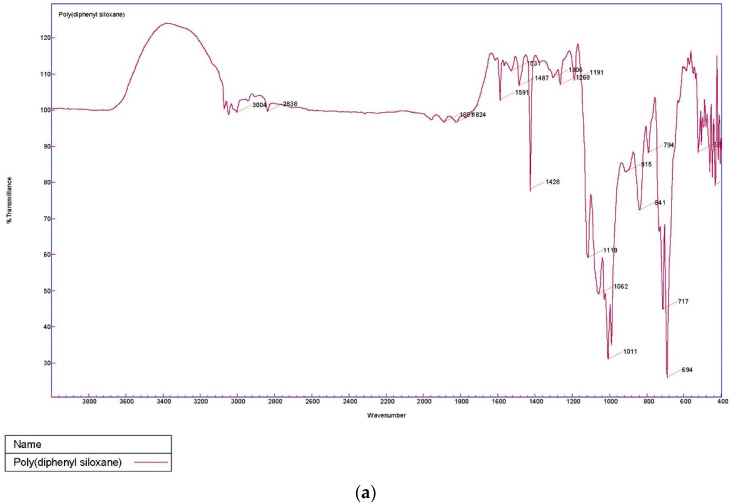
Fourier Transform Infrared Spectroscopy (FT-IR) spectra of (**a**) pristine poly (diphenylsiloxane) (PDPS) polymer and (**b**) sol–gel PDPS coated fiberglass.

**Figure 3 molecules-26-00009-f003:**
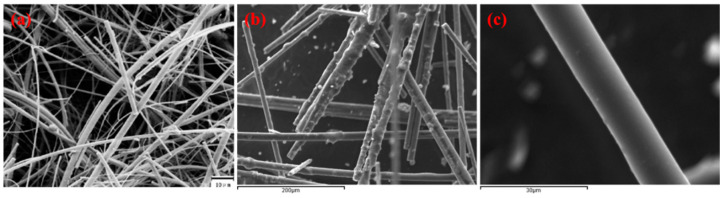
Scanning Electron Microscopy images of (**a**) uncoated glass fiber; (**b**) sol–gel PDPS coated glass fibers; (**c**) magnified image of sol–gel PDPS coated individual glass fiber.

**Figure 4 molecules-26-00009-f004:**
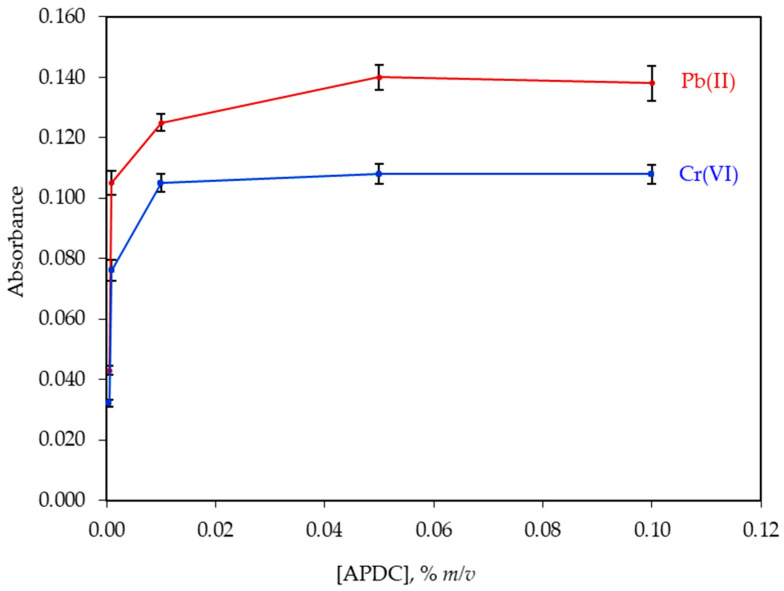
Effect of ammonium pyrrolidine dithiocarbamate (APDC) concentration on the absorbance of 100.0 μg·L^−1^ Pb(II) and 100.0 μg·L^−1^ Cr(VI). All other experimental parameters as presented in Table 1. Error bars were calculated based on standard deviation values (*n* = 5).

**Figure 5 molecules-26-00009-f005:**
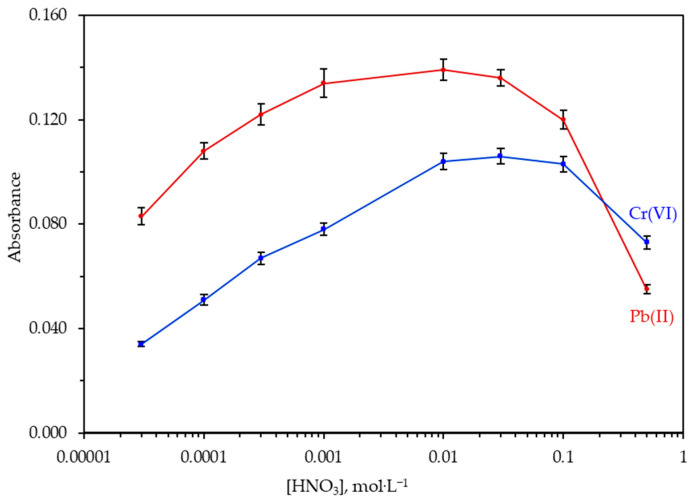
Effect of sample acidity on the absorbance of 100.0 μg·L^−1^ Pb(II) and 100.0 μg·L^−1^ Cr(VI). All the other experimental parameters are as presented in Table 1. Error bars were calculated based on standard deviation values (*n* = 5).

**Table 1 molecules-26-00009-t001:** Operational sequences of the FI-FCSE-FAAS system for metal determination.

Step	Valve Position		Pump Operation		Delivered Medium	Flow Rate (mL·min^−1^)	Time (s)	Operation
	IV	V	P	SP				
1	Load	R1	ON	OFF	Sample or Std	8.0	60	Preconcentration
					0.05% (*m*/*v*) APDC	1.0		
	Load	R1	OFF	Aspirate	MIBK	9.0		Aspiration 1500 μL MIBK
2	Elution	R2	OFF	Dispense	MIBK	3.0	30	Elution/measurement

**Table 2 molecules-26-00009-t002:** Analytical features of the FI-FCSE-FAAS method for lead and chromium(VI) determination.

Parameter	Chromium(VI)		Lead	
Preconcentration time (s)	60	120	60	120
Enhancement factor	47	70	150	215
Linear range (μg·L^−1^)	6.9–350	4.2–200	5.3–200	3.8–200
Sensitivity (Slop), (μg·L^−1^)	0.0011 ± 0.5 × 10^−4^	0.0018 ± 0.9 × 10^−4^	0.0014 ± 0.5 × 10^−4^	0.0020 ± 1.1 × 10^−4^
Detection limit (3 s) (μg·L^−1^)	2.1	1.2	1.6	1.1
Quantification lim. (10 s) (μg·L^−1^)	6.9	4.2	5.3	3.8
Precision (RSD, *n* = 10) (%)	3.3 (30.0 μg·L^−1^)	3.2 (20.0 μg·L^−1^)	2.8 (30.0 μg·L^−1^)	3.0 (20.0 μg·L^−1^)
Correlation coefficient (*r*)	0.9987	0.9997	0.9994	0.9984

**Table 3 molecules-26-00009-t003:** Comparisons of the analytical performance of the present method with other on-line SPE-FAAS methods reported in the literature for lead and chromium(VI) determination. SPE: solid phase extraction.

Analyte	Sorbent Material	Reagent	Eluent	PT (s)	SC (mL)	LOD (μg·L^−1^)	RSD (%)	EF	Ref.
Pb	Sol–gel PDMDPS polyester fabric disks	APDC	MIBK	90	15.6	1.8	3.1	140	[16]
	Sol–gel PDMS fabric fiber	DDTC	MIBK	90	18	1.6	2.9	167	[17]
	Oasis^®^-HLB	DDTP	Methanol	90	12.0	0.92	2.6	180	[24]
	HyperSepSCX resin	-	2 mol·L^−1^ HCl	150	15.0	2.1	3.1	97	[25]
	Bond Elut Plexa PCX resin	-	1 mol·L^−1^ HCl	90	18.0	1.8	3.1	95	[26]
	Amberlite XAD-4 functionalized with 2,6-pyridinedicarboxaldehyde	-	1 mol·L^−1^ HNO_3_	660	10	2.2	-	27.9	[27]
	Nobias chelate PA-1	-	1.5 mol·L^−1^ HNO_3_	120	20.0	1.6	3.3	98	[28]
	Strata^TM^-X resin	DDTC	Methanol	90	15.6	1.6	2.9	140	[29]
	GF-Sol–gel PDPS	APDC	MIBK	120	16.0	1.1	3.0	215	Present work
Cr	Strata^TM^-X resin	DDTC	Methanol	90	15.6	1.2	3.8	63	[29]
	Oasis HLB	DDTC	Methanol	90	10	0.8	3.5	70	[30]
	PTFE beads	Thio-semicarbazide	0.5 mol·L^−1^ HNO_3_	120	10.0	0.14	1.3	56	[31]
	Animal fiber	-	1.0 mol·L^−1^ NaOH	750	25.0	0.3	4.3	32	[32]
	PS-NAPdien	-	BufferNH_3/_NH_4_NO_3_	150	10.0	2.5	0.8	30	[33]
	Dowex 21K resin	-	3 mol·L^−1^ HNO_3_	204	5.8	0.3	4.0	30	[34]
	SiO_2_-Zr	-	THAM	300	20.0	2.3	2.1	25	[35]
	GF-Sol–gel PDPS	APDC	MIBK	120	16.0	1.2	3.2	77	Present work

PT: preconcentration time, SC: sample consumption, RSD: relative standard deviation, LOD: limit of detection, EF: enrichment factor, GF: Sol–gel PDPS: glass fiber coated with sol–gel poly-diphenylsiloxane, PS-NAPdien: chloromethylated polystyrene functionalized with *N*,*N*-bis(naphthylideneimino)diethylenetriamine, THAM: tris(hydroxymethyl)methylamine.

**Table 4 molecules-26-00009-t004:** Analytical results for lead and chromium(VI) determination in CRMs.

Analyte	CRM	Units	Certified Value	Found *	Relative Error (%)	*t* _exp_
Pb	NIST 1643e	μg·L^−1^	19.63 ± 0.21	18.5 ± 0.6	5.8	3.262
	BCR 278-R	mg·kg^−1^	2.00	2.10 ± 0.06	−5.0	−2.887
Cr(VI)	SRM 2109	μg·L^−1^	20.0	19.2 ± 0.5	4.0	2.771

* mean value ± standard deviation based on three replicates; *t*_crit_ = 4.303 at 95% probability level.

**Table 5 molecules-26-00009-t005:** Application of the FI-FCSE-FAAS method for lead and chromium(VI) determination in spiked natural water samples.

Analyte	Sample Type	Added ^a^	Found ^a^	*R* (%)
Pb(II)	Axios river water	-	3.5 ± 0.1	
		20.0	23.2 ± 0.7	98.5
	Volvi lake water		N.D.	
		20.0	28.8 ± 0.8	94.0
	Toroneos gulf seawater		N.D.	
		20.0	21.6 ± 0.6	108.0
Cr(VI)	Axios river water	-	5.9 ± 0.2	
		20.0	26.5 ± 0.8	103.0
	Volvi lake water	-	N.D.	
		20.0	19.5 ± 0.5	97.5
	Toroneos gulf seawater	-	N.D.	
		20.0	18.6 ± 0.9	93.0

^a^ Concentrations in μg·L^−1^, mean value ± standard deviation; N.D., not detected.

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
