# Peer review of "A Novel Glass Fiber Coated with Sol–Gel Poly-Diphenylsiloxane Sorbent for the On-Line Determination of Toxic Metals Using Flow Injection Column Preconcentration Platform Coupled with Flame Atomic Absorption Spectrometry"

_molecules, 2020, doi:10.3390/molecules26010009_

Round 1

Reviewer 1 Report

Comments and Suggestions for Authors

The authors describe in this manuscript a novel material for the preconcentration of toxic metal at trace concentration based on flow injection column preconcentration and flame atomic absorption spectrometry. The manuscript has sufficient novelty, quality and interest for the readership of ‘Molecules’ to be published.

It is remarkable how the authors manage to use preconcentration with a newly prepared sol-gel based adsorbent to preconcentrate lead and chromium ions to that extent that even with the relatively insensitive flame-AAS as detection technique LODs in the low ppb-range can be reached. I was particularly intrigued by the possibility to assess the concentration of the (far more toxic) Cr(VI) with the APDC chelating agent through in situ-reduction, rather than to determine the (relatively non-toxic) Cr(III) as would normally have been expected.

There are a number of smaller items which would probably need correction, but these are only of minor / technical / lingual character and should not hinder its publication.

These are:

Abstract, l.15: potential, not potentials

Abstract, l.16: consequently then also: “was demonstrated” (instead of “were demonstrated”)

Abstract, l.24: “tolerance to” or “tolerance of” instead of “tolerance on”

Introduction, l.35: “successive” or “successful” ?

Introduction, l.37: No comma after “extraction”

Introduction, l.48: I don’t see the clear difference between “hydrophobic” and “reversed phase” (mechanisms) – please clarify !

Introduction, l. 64: An excellent review to be quoted instead or in addition to ref. [10] in the particular context is the one from Z. Mester and R.E. Sturgeon, Trace element speciation using solid phase microextraction. Spectrochimica Acta Part B Atomic Spectroscopy 60(9) (2005) 1243-1269.

Introduction, l. 76: “in an on line …”

Introduction, l. 79: clarify or rephrase: “sol-gel coated derived polyester fibers”

Introduction, l. 80: typo: front end

Materials and Methods, l.122: Rephrase: “The detection system consisted of…”

Materials and Methods, l.139: include -> includes. Also: is the 5-port valve a commercially available one ? How is this realized in practice ? I assume that the Figure 1 is only a schematic representation of the 5-port valve. Could this be drawn in Figure 1 a bit closer to its actual look ?

Materials and Methods, l.158: “scissors” is a plural-only word !

Materials and Methods, l.176: plural: two fiber types

Results and Discussion, l. 252: Sentence is unclear and must be rephrased: “as (more ?) effective eluents than acidic or alkaline solutions”

Results and Discussion, l. 260: Sentence must be rephrased: “The most commonly used chelating agents…”

Figure 2: Have the results presented in this graph been measured in replicates ? If so, please include error bars.

Figure 3: Same comment as above.

Results and Discussion, l. 322: There is no reference to Figure 4 in the manuscript, but there is one to Figure 5. However, neither Figure 4 nor 5 are included in the main body of the manuscript. Did these get lost ?

Results and Discussion, l. 339: “are present” , not “are presented” !

Results and Discussion, l. 363: “…because cotton is a hydrophilic substrate” – is that still true after the surface of the cotton fibres has been derivatized and thus been made more (or rather: completely) hydrophobic ?

Conclusion, l.414: Rephrase: “It is worth being noted”. Also: Rephrase the entire sentence, as it is difficult to understand. Also rephrase: “The resulting technique…”

References: Check author list of ref. [12].

Author Response

Response to Reviewer 1 Comments

The authors describe in this manuscript a novel material for the preconcentration of toxic metal at trace concentration based on flow injection column preconcentration and flame atomic absorption spectrometry. The manuscript has sufficient novelty, quality and interest for the readership of ‘Molecules’ to be published.

It is remarkable how the authors manage to use preconcentration with a newly prepared sol-gel based adsorbent to preconcentrate lead and chromium ions to that extent that even with the relatively insensitive flame-AAS as detection technique LODs in the low ppb-range can be reached. I was particularly intrigued by the possibility to assess the concentration of the (far more toxic) Cr(VI) with the APDC chelating agent through in situ-reduction, rather than to determine the (relatively non-toxic) Cr(III) as would normally have been expected.

There are a number of smaller items which would probably need correction, but these are only of minor / technical / lingual character and should not hinder its publication.

These are:

  1. Abstract, l.15: potential, not potentials

Response: We agree, it has been changed in the revised MS.

  1. Abstract, l.16: consequently then also: “was demonstrated” (instead of “were demonstrated”)

Response: We agree, it has been changed, in the revised MS.

  1. Abstract, l.24: “tolerance to” or “tolerance of” instead of “tolerance on”.

Response: We agree, it has been changed, in the revised MS.

  1. Introduction, l.35: “successive” or “successful”?

Response: It has been changed as “successful”, in the revised MS.

  1. Introduction, l.37: No comma after “extraction”.

Response: We agree, it has been changed, in the revised MS.

  1. Introduction, l.48: I don’t see the clear difference between “hydrophobic” and “reversed phase” (mechanisms) – please clarify!

Response: Actually, hydrophobic and reversed phase are the same mechanism. The sentence has been rephrased as: “…….. as hydrophobic (reversed phase), polar and ion-exchange.”, in the revised MS.

  1. Introduction, l. 64: An excellent review to be quoted instead or in addition to ref. [10] in the particular context is the one from Z. Mester and R.E. Sturgeon, Trace element speciation using solid phase microextraction. Spectrochimica Acta Part B Atomic Spectroscopy 60(9) (2005) 1243-1269.

Response: The authors have carefully reviewed the article suggested by the Reviewer. It was found to be relevant to the current manuscript and quite comprehensive. As such, a relative sentence for this reference has been added at the end of the 2nd paragraph of the revised manuscript.

  1. Introduction, l. 76: “in an on line …”.

Response: We agree, it has been changed.

  1. Introduction, l. 79: clarify or rephrase: “sol-gel coated derived polyester fibers”

Response: We rephrased the sentence, in the revised MS.

  1. Introduction, l. 80: typo: front end.

Response: We agree, it has been changed, in the revised MS.

  1. Materials and Methods, l.122: Rephrase: “The detection system consisted of…”

Response: We agree, it has been changed, in the revised MS.

  1. Materials and Methods, l.139: include -> includes.

Response: We agree, it has been changed, in the revised MS.

  1. Also: is the 5-port valve a commercially available one? How is this realized in practice? I assume that the Figure 1 is only a schematic representation of the 5-port valve. Could this be drawn in Figure 1 a bit closer to its actual look?

Response: The 5-port injection valve is a part of the FIAS-400 flow injection system (Perkin-Elmer), which is commercially available, as it is presented in 2.2. Apparatus section. A close-up photo of the 5-port injection valve is given in supplementary material Figure S2. On the other hand, Figure 1 is only a schematic diagram of the manifold used in the method. We agree with the Reviewer’s suggestion for a new drawn in Figure 1 closer to its actual look. As such, we redesigned Figure 1, in the revised MS

  1. Materials and Methods, l.158: “scissors” is a plural-only word!

Response: We agree, it has been changed, in the revised MS.

  1. Materials and Methods, l.176: plural: two fiber types.

Response: We agree, it has been changed, in the revised MS.

  1. Results and Discussion, l. 252: Sentence is unclear and must be rephrased: “as (more ?) effective eluents than acidic or alkaline solutions”

Response: We rephrased the sentence as “……. various organic solvents as well as acidic or alkaline solutions can be used for the effective elution of the extracted analytes without risking the potential loss of the extracting sorbent.” in order to be clearer.    

  1. Results and Discussion, l. 260: Sentence must be rephrased: “The most commonly used chelating agents…”.

Response: We agree, it has been changed, in the revised MS

  1. Figure 2: Have the results presented in this graph been measured in replicates? If so, please include error bars.

Response: Five replicate measurements per value were performed in all instances. As such, error bars were calculated based on standard deviation values and included in the graph.

  1. Figure 3: Same comment as above.

Response: Error bars included in the graph, in the revised MS.

  1. Results and Discussion, l. 322: There is no reference to Figure 4 in the manuscript, but there is one to Figure 5. However, neither Figure 4 nor 5 are included in the main body of the manuscript. Did these get lost?

Response: Figure 4 and 5 are presented in the supplementary materials defined as Figure S4 and S5.

  1. Results and Discussion, l. 339: “are present” , not “are presented”!

Response: We agree, it has been changed, in the revised MS

  1. Results and Discussion, l. 363: “…because cotton is a hydrophilic substrate” – is that still true after the surface of the cotton fibres has been derivatized and thus been made more (or rather: completely) hydrophobic?

Response: As the sol gel PDPS coated is hydrophobic agent the hydrophilic cotton fibers expected to derivatized in a hydrophobic sorbent. This is true. But as we observed, gradually tightening of cotton fibers into the column resulting in high backpressure, we attributed this problem probably to thinner fibers and the hydrophilic nature of cotton. As such, we rewritten the two last sentences of the paragraph.

  1. Conclusion, l.414: Rephrase: “It is worth being noted”. Also: Rephrase the entire sentence, as it is difficult to understand. Also rephrase: “The resulting technique…” Response: We agree, and we rephrase the two sentences, in the revised MS.

  1. References: Check author list of ref. [12].

Response: We checked carefully the names of the authors. These are right. No changed have been done.

Reviewer 2 Report

This manuscript includes interesting data. Therefore, I basically recommend publication of this work in Materials. However, some revisions are necessary. Please see below.

1) This manuscript provides lots of function data. However data on materials characterizations are very poor. Please provide standard data of the used sol-gel materials (IR, SEM, specific surface area etc)

2) References are not well updated and generalized. Recent progresses of materials for separation and detection had better be described shortly with citing related papers (for example, see below).

https://www.journal.csj.jp/doi/abs/10.1246/bcsj.20190372

https://onlinelibrary.wiley.com/doi/abs/10.1002/elps.201800331

https://www.journal.csj.jp/doi/abs/10.1246/bcsj.20180280

https://www.sciencedirect.com/science/article/abs/pii/S0039914019303546?via%3Dihub

3) Some paragraphs are too short. Please combine too short paragraphs into one paragraph with appropriate length.

Author Response

Response to Reviewer 2 Comments

This manuscript includes interesting data. Therefore, I basically recommend publication of this work in Materials. However, some revisions are necessary. Please see below.

1) This manuscript provides lots of function data. However, data on materials characterizations are very poor. Please provide standard data of the used sol-gel materials (IR, SEM, specific surface area etc)

Response: 

The authors do agree with the Reviewer’s comment about the characterization of sol-gel sorbent used in the manuscript. As such, new characterization data using FT-IR and Scanning Electron Microscopy have been added to the revised manuscript.

2) References are not well updated and generalized. Recent progresses of materials for separation and detection had better be described shortly with citing related papers (for example, see below).

https://www.journal.csj.jp/doi/abs/10.1246/bcsj.20190372

https://onlinelibrary.wiley.com/doi/abs/10.1002/elps.201800331

https://www.journal.csj.jp/doi/abs/10.1246/bcsj.20180280

https://www.sciencedirect.com/science/article/abs/pii/S0039914019303546?via%3Dihub

Response: 

The authors have carefully reviewed the articles suggested by the Reviewer. Only the first article was found relevant to the current manuscript. As such, it has been included in the revised manuscript (1. Introduction, first paragraph).   

 3) Some paragraphs are too short. Please combine too short paragraphs into one paragraph with appropriate length.

Response:

We agree, and we thank the Reviewer for his comment. We have combined several short paragraphs into one in the revised manuscript. 

Reviewer 3 Report

In this work  glass fiber coated with sol-gel poly-2 diphenylsiloxane sorbent was prepared for the on-line determination of toxic metals by using flow injection column preconcentration platform coupled with FAAS. The platform was applied to trace lead and chromium(VI) in environmental water samples and verified by analyzing certified reference materials and spiked environmental water samples. The developed system may be useful for automated determination of metals at low concentration, however, some issues need to be clarified in more depth:

- in sub-section „2.4. Preparation of glass fiber and cotton ball coated with sol-gel sorbents”

-- please provide characteristics of the glass fiber (type, kind of functionalization – if any, length, aspect ratio) and the cotton ball;

--   „The segments of fiber substrates (e.g., glass, cotton) are first soaked (…) Then, fibers are cleaned (…) After, a process called mercerization follows, by treating the fibers with 1.0 mol L-1 NaOH under sonication…”  -please explain how mercerization process works with glass fibres?

- Authors applied methyltrimethoxysilane which  systems  are  prone  to  phase  separation  due  to  the hydrophobic nature of Si-CH3 groups, low degree of cross-linking  (f=3),  and  weak  mechanical  properties – how those drawbacks were avoided?

- please provide structural (spectroscopic) and morphological (SEM) characteristics of the coating on the obtained  coated glass fibers/cotton balls; what about structural deformations that may cause  internal stress?

- what is the porosity of these coatings?

- what is the reproducibility of the sol-gel procedure?

- please refer to the stability of the developed coatings over time and multiple extraction processes.

Author Response

Response to Reviewer 3 Comments

In this work glass fiber coated with sol-gel poly-2 diphenylsiloxane sorbent was prepared for the on-line determination of toxic metals by using flow injection column preconcentration platform coupled with FAAS. The platform was applied to trace lead and chromium(VI) in environmental water samples and verified by analyzing certified reference materials and spiked environmental water samples. The developed system may be useful for automated determination of metals at low concentration, however, some issues need to be clarified in more depth:

1) in sub-section, 2.4. Preparation of glass fiber and cotton ball coated with sol-gel sorbents”:

  1. a) please provide characteristics of the glass fiber (type, kind of functionalization – if any, length, aspect ratio) and the cotton ball;

Response:

The fiber glass is made from fine glass fibers with diameter of individual filament approximately 0.008 mm. The fiber glass is pliable and resistant to breakage. The borosilicate glass used for preparing the fiber glass contains minimal heavy metals-an attribute advantageous for heavy metal extraction and analysis. The fiber glass contains high concentration of surface Si-OH functional groups that can be easily exploited to anchor with ligand/polymer of choice to create a new microextraction system via sol-gel sorbent coating technology.

The cotton ball is made from 100% cotton cellulose possessing hydrophilic surface property due to the presence of accessible C-OH functionality. These Carbinol functional groups can be used to chemically bind different polymer/ligand via sol-gel process to create new microextraction system.

These information has been added in the revised manuscript, section: 2.4. Preparation of glass fiber and cotton ball coated with sol-gel sorbents.

  1. b) The segments of fiber substrates (e.g., glass, cotton) are first soaked (…) Then, fibers are cleaned (…) After, a process called mercerization follows, by treating the fibers with 1.0 mol L-1 NaOH under sonication…” -please explain how mercerization process works with glass fibres?

Response:

The authors sincerely appreciate the thoroughness of the Reviewer. The term “Mercerization” is applicable to cellulose substrate to clean the substrate and to reorient the structure so that the substrate becomes stronger and cleaner for the downstream sol-gel sorbent coating. The same treatment regimen is also applied to the fiber glass that increases the available surface silanol groups (Si-OH), although it cannot be called mercerization.         

2) Authors applied methyltrimethoxysilane which systems are prone to phase separation due to the hydrophobic nature of Si-CH3 groups, low degree of cross-linking (f=3), and weak mechanical properties – how those drawbacks were avoided?

Response:

The Reviewer has made an excellent point. The low degree of cross-linking results in weak mechanical strength when the coating is thick or monolithic in nature. However, the sol-gel sorbent coating used in the current study was ultra-thin (sub micrometer) with relatively open structure compared to tetramethoxysilane. Due to the open structure, the aqueous solution may easily access to the poly(diphenylsiloxane) polymer implanted into the polymeric network.     

3) please provide structural (spectroscopic) and morphological (SEM) characteristics of the coating on the obtained coated glass fibers/cotton balls; what about structural deformations that may cause internal stress?

Response:    

In response to the Reviewer’s suggestion, the authors have added new characterization data using FT-IR and Scanning Electron Microscopy in the revised manuscript.

Due to the ultra-thin sol-gel sorbent coating around the individual fiber glass filament/cellulose micro fibril, there is no substantial internal stress on the sorbent coated fiber glass/cotton ball. However, the original fluffiness of the substrates is lost to some extent after the sol-gel sorbent coating.

4) what is the porosity of these coatings?

Response:

The sol-gel sorbent is coated on the surface of individual fiber glass and therefore the bulk fiber glass mass retains its permeability after the sol-gel sorbent coating. As a result, when the aqueous solution of heavy metals is passed through the bed, the solution passes through it with minimal resistance. In addition, sol-gel sorbents are inherently porous (consisting of micro and mesopores). The porous sol-gel sorbent coating substantially increases the overall surface area/unit mass of the fiber glass and allows easy access to the implanted poly(diphenylsiloxane) polymers for interacting with the target metal species.         

5) what is the reproducibility of the sol-gel procedure?

Response:

Unlike sorbent coating process used in classical microextraction techniques, sol-gel sorbent coating process is a highly reproducible chemical coating process. By precise control of the sol solution formulation and the coating temperature (ambient/ near ambient), the sol-gel sorbent can be reproduced with precise coating thickness and sorbent loading. The reproducibility of the sol-gel sorbent loading always remain within 0-5% RSD.     

6) please refer to the stability of the developed coatings over time and multiple extraction processes.

Response: 

This information is already given in Section 2.3. Construction of the FCSE microcolumn, line-170, in the initial manuscript; “….. the retention efficiency was unchanged as well as the packing material remained stable for more than 500 loading/elution cycles for the two metals.”   

Round 2

Reviewer 3 Report

The revised manuscript can be published as it stands. Authors provided responses to all points raised by the reviewer.